# Generation of a Dystrophin Mutant in Dog by Nuclear Transfer Using CRISPR/Cas9-Mediated Somatic Cells: A Preliminary Study

**DOI:** 10.3390/ijms23052898

**Published:** 2022-03-07

**Authors:** Hyun Ju Oh, Eugene Chung, Jaehwan Kim, Min Jung Kim, Geon A. Kim, Seok Hee Lee, Kihae Ra, Kidong Eom, Soojin Park, Jong-Hee Chae, Jin-Soo Kim, Byeong Chun Lee

**Affiliations:** 1Department of Theriogenology and Biotechnology, College of Veterinary Medicine, Seoul National University, Seoul 08826, Korea; newborn52020@gmail.com (H.J.O.); tinia19@snu.ac.kr (M.J.K.); pshsje03@snu.ac.kr (G.A.K.); tempuler@snu.ac.kr (S.H.L.); ragh1102@naver.com (K.R.); 2Center for Genome Engineering, Institute for Basic Science, Seoul 08826, Korea; eugene1990@snu.ac.kr; 3Department of Chemistry, Seoul National University, Seoul 08826, Korea; 4Department of Veterinary Medical Imaging, College of Veterinary Medicine, Konkuk University, Seoul 5029, Korea; jaehwan@konkuk.ac.kr (J.K.); eomkd@konkuk.ac.kr (K.E.); 5Department of Clinical Pathology, College of Health Science, Eulji University, Uijeongbu 11759, Korea; 6Department of Pediatrics, Seoul National University College of Medicine, Seoul 03080, Korea; sj8383@snu.ac.kr (S.P.); chaeped1@snu.ac.kr (J.-H.C.)

**Keywords:** dystrophin, mutant, CRISPR/Cas9, dog, somatic cell nuclear transfer

## Abstract

Dystrophinopathy is caused by mutations in the dystrophin gene, which lead to progressive muscle degeneration, necrosis, and finally, death. Recently, golden retrievers have been suggested as a useful animal model for studying human dystrophinopathy, but the model has limitations due to difficulty in maintaining the genetic background using conventional breeding. In this study, we successfully generated a dystrophin mutant dog using the CRISPR/Cas9 system and somatic cell nuclear transfer. The dystrophin mutant dog displayed phenotypes such as elevated serum creatine kinase, dystrophin deficiency, skeletal muscle defects, an abnormal electrocardiogram, and avoidance of ambulation. These results indicate that donor cells with CRISPR/Cas9 for a specific gene combined with the somatic cell nuclear transfer technique can efficiently produce a dystrophin mutant dog, which will help in the successful development of gene therapy drugs for dogs and humans.

## 1. Introduction

Canis familiaris has drawn considerable attention as a model for investigating human diseases. Dogs show over 450 naturally occurring diseases, of which approximately 360 are analogous to human diseases [1]. Based on their size, biological features, and ease of behavioral evaluation and handling, dogs can be good animal models. Since humans and dogs share a common environment, food, and carcinogenic load, it is not surprising that the dog has emerged as a viable model for human disease [2]. Duchenne muscular dystrophy (DMD) and Becker muscular dystrophy (BMD) are the most common X-linked recessive muscular dystrophies caused by mutations in the dystrophin gene leading to a defective dystrophin–glycoprotein complex [3]. DMD and BMD are classified as dystrophinopathies because they are caused by alterations in the dystrophin gene. These mutations lead to progressive muscle degeneration and, finally, to necrosis [3,4]. This can result in substantial physical and locomotor deficits, leading to the need for wheelchair use and early death due to heart failure. Both laboratory-generated and naturally occurring animal models are available to study the pathogenesis of dystrophinopathy and to develop potential new treatments [5]. Dystrophin-deficient mdx mice have been most commonly used for DMD research, but this model has limitations. For example, mdx mice exhibit minimal clinical symptoms and have only a 25% reduction in longevity, unlike DMD patients, who have a 75% reduction in life span [6]. In addition, there is a weak correlation between the effect of therapeutic interventions in the rodent model and the effect observed in humans [7]. Thus, rodents are not good models of human dystrophinopathy. Recently, canine models have been suggested as a suitable translational bridge between mice and humans [8,9] because they more closely mimic the human disease compared with other existing models of dystrophin deficiency [10].

Canine X-linked muscular dystrophy models have been reported over the last 50 years. Generally, the clinical phenotype of canine dystrophinopathy is more similar to that of human patients in severity and in selective muscle injury compared with mdx mice [9]. Overall, golden retriever muscular dystrophy (GRMD) has been the most extensively examined and characterized for research on human DMD [11]. Mutations in the canine dystrophin gene have been identified in golden retrievers [12], German short-hair pointers [13], and Cavalier King Charles spaniels [14]. However, GRMD dogs have a high degree of variability despite having an identical causative mutation, leading to a large phenotypic range resulting from variation in the alternatively spliced dystrophin gene and truncated translational products in the muscles. Additionally, using large golden retrievers has significant ethical consequences regarding animal welfare, and their maintenance and care are expensive. To address these issues, the GRMD model was bred with a beagle, and a new colony with canine X-linked muscular dystrophy (CXMD) was generated [15]. In addition to the clinical resemblance, the CXMD model also shows histological lesions similar to those seen in affected humans. However, it is difficult to produce individuals with the same genetic background as the CXMD model using conventional breeding methods and maintain individuals for use in preclinical studies. To increase the availability of canine dystrophinopathy models, a corresponding loss-of-function model using site-directed mutagenesis of the desired gene is needed. 

Recently, the clustered regularly interspaced short palindromic repeats (CRISPRs)/CRISPR-associated (Cas) 9 system was developed to edit specific genes with high efficiency [16,17]. Using this technique, numerous genome-edited animals from different species have been generated for biomedical modeling [18,19,20,21], human disease modeling [22,23,24], and xenotransplantation [25,26,27]. In fact, apoE knockout (KO) dogs [28] and myostatin KO dogs [29] were successfully produced by microinjecting CRISPR/Cas9 into zygotes. Although zygote microinjection may be very highly efficient in editing, not all of the resulting embryos are true KOs, and the approach can generate mosaic embryos. Somatic cell nuclear transfer (SCNT) is currently the only technique to ensure that all fetuses are KOs because the experimenter can select the donor cell after establishing donor cells that are edited with the desired genes before the SCNT procedure. In this study, a dystrophin mutant dog was successfully generated using the CRISPR/Cas9 system combined with SCNT technology.

## 2. Results

### 2.1. Generation of Dystrophin Mutant Cloned Dog by SCNT

The efficiency of sgRNA was validated by co-transfection with a Cas9 vector into canine male fetal fibroblasts by T7E1 assay (Appendix A). Having confirmed that Cas9/sgRNA was highly active in cultured cells, we performed SCNT to generate a dystrophin mutant dog. In total, 49 in vivo matured oocytes from 4 oocyte donor dogs were recovered, and these oocytes were enucleated, injected with a donor cell, and fused by electrical stimulation. The fused couplets (26/49, 53.1%) were activated with 10 μM calcium ionophore and 6-dimethylaminopurine, and then 26 reconstructed oocytes were transferred into three naturally synchronized surrogate recipients. One of the recipients was pregnant to term (33.33%) and gave birth to one puppy (3.85%) (Table 1).

The tail tissue of the cloned puppy was collected two days after birth to detect genomic mutations in the target dystrophin locus using the T7E1 assay and deep sequencing. PCR products amplified from the genome of this dog were identified via deep sequencing (Appendix A). As shown in Figure 1A, the puppy had a 57 bp deletion in the dystrophin gene (Figure 1A). The off-target sites in the dog genome were identified using Cas-OFFinder [30]. No off-target indel mutations at candidate sites were detected (Appendix A).

### 2.2. Creatine Kinase and Electrocardiographic Analysis

The CK level was recorded in both the dystrophin mutant dog and an age-matched control at two and eight weeks after birth. Until eight weeks of age, the control group had CK levels within the normal range (99–436 U/L), but the CK level (1019–19,880 U/L) of the dystrophin mutant dog was much higher than the normal range from two weeks to eight weeks (Figure 1B). At 10 months of age, the CK levels of the dystrophin mutant dog and the control dog were 261 and 31,540 U/L, respectively. However, after 30 min of exercise, the dystrophin mutant dog had a CK level 300-fold higher than that of the control group (Figure 1C). The electrocardiogram (ECG) patterns of a five-month-old dystrophin mutant dog and the age-matched control were similar. However, the Q wave in the ECG was deeper in the five-month-old dystrophin mutant dog than in the age-matched control (Appendix A).

### 2.3. Muscle Magnetic Resonance Imaging

For profiling skeletal muscles, magnetic resonance imaging (MRI) of the five-month-old dystrophin mutant dog was performed (Figure 2). Diffuse hyperintense lesions on T2-weighted and T2 fat suppression sequences were found, especially in the rectus femoris and adductor magnus muscles. On T1-weighted images, atrophy of the quadriceps and strong contrast enhancement were found in the rectus femoris muscle. The biceps femoris, semitendinosus, and semimembranosus muscles tended to be relatively uninvolved. T2 values were acquired using T2 mapping of both control and dystrophin mutant dogs (Appendix A). The average T2 values of the dystrophin mutant dog were much higher (mean ± SD, 45.8 ± 8.2) than those of the control dog (mean ± SD, 38.7 ± 1.9). Significant differences between the two dogs were found in anterior-medial hind limb muscles, including the rectus femoris (control dog, 38.1; dystrophin mutant dog, 62.7) and adductor magnus (control dog, 40.3; dystrophin mutant dog, 49.3) muscles. In contrast, the posterior-lateral hind limb muscles, including the biceps femoris, semitendinosus, and semimembranosus, showed minimal differences between the two dogs.

### 2.4. Histopathological Analysis

The present study examined the biceps femoris of the dystrophin mutant dog and a control dog at six months of age. Muscle histopathology examination revealed mild fiber size variations, muscle fiber necrosis, and regeneration in focal muscle groups (Figure 3). Immunohistochemical staining of frozen muscle using monoclonal antibodies against the dystrophin carboxy-terminal domain, rod domain, and utrophin revealed decreased expression of dystrophin 1 and 2 (Figure 3F,G) along with the upregulation of utrophin (Figure 3H), compared to the control dog muscles (Figure 3A–D).

### 2.5. Western Blotting

Dystrophin 1 and dystrophin 2 were detected in the muscle tissue of the control in Western blots but were barely expressed in the dystrophin mutant dog (Figure 4). In the Western blot, utrophin is markedly upregulated in the dystrophin mutant dog when compared with controls (Figure 4).

## 3. Discussion

In the present study, we demonstrated that a dystrophin mutant dog could be generated by nuclear transfer using donor cells in which the dystrophin gene has been knocked out by the CRISPR/Cas9 system. First, we designed a sgRNA targeting exon 6 of the dog dystrophin gene [31]. The remaining approximately 7% of dystrophin gene mutations are caused by single- or multi-exon duplications, with exons 2 to 20 being the most commonly affected. It is important to note that the GRMD and CXMD models have a frame shift because of a mutation in the splice acceptor of exon 6 that disrupts exon 7, and inducing a mutation in exon 6 could have a therapeutic effect on these canine dystrophinopathy models [32]. Therefore, the present study successfully produced dogs with exon 6 mutations by SCNT.

We next evaluated whether the dystrophin mutant dog shared a remarkably similar clinical course to that of dystrophinopathy patients. In both GRMD dogs and human DMD patients, serum CK activity is markedly elevated [33]. CK is the most important for pre-neuter evaluations in young dogs because increased CK activity may be an early indicator of underlying muscle disease. With this in mind, we observed the clinical manifestations of dystrophinopathy in the dystrophin mutant dog with increased CK using MRI. 

On MRI imaging, significant differences between the control and dystrophin mutant dog were found in anterior-medial hind limb muscles, including the rectus femoris and adductor magnus muscles, and the dystrophin mutant dog showed diffuse hyperintense lesions. These MRI findings are similar to those of patients with dystrophinopathy, especially young boys with DMD, regarding the existence of marked inflammatory and edematous lesions with minimal fatty replacement in the thigh muscles [34,35]. Since the MRI distribution and patterns were very similar to those of early-stage DMD boys, it is likely that muscular atrophy and fatty infiltration occur as the disease progresses. 

To establish the dystrophin mutant dog as a valid dystrophinopathy model, we performed ECG analysis in an age-matched control and the dystrophin mutant dog. Patients with dystrophinopathy often show an abnormal ECG. It is generally believed that end-stage patients develop dilated cardiomyopathy and die from heart failure [36]. Such ECG changes are well established in human dystrophinopathy, but few studies have been reported in dystrophin-deficient dogs [37,38]. In our results, there were no differences in ECG patterns between the dystrophin mutant dog and control, but age-related changes will require follow-up. 

In addition, we observed walking- or exercise-related changes in the dystrophin mutant dog at five months of age (data not shown). The dystrophin mutant dog started to show “bunny hopping” from six months of age. Over time, its limbs became stiff, with a decreased range of joint motion while moving, more pronounced bunny hopping, difficulty in climbing stairs, and avoidance of movement. In untreated DMD patients, ambulation loss usually occurs during the early teenage years. Unlike in DMD patients, complete loss of ambulation is not a clinical feature in young DMD dogs [8,10]. In this study, the 10-month-old dystrophin mutant dog was reluctant to exercise and showed limb muscle atrophy but was still able to walk (data not shown). Because of the clinical symptoms seen in DMD, it is necessary to observe whether the dystrophin mutant dog will completely lose ambulation in the future. 

In order to observe the muscle phenotype, the present study examined histopathological analysis using the biceps femoris of the dystrophin mutant dog and a control dog at six months of age. In immunohistochemical staining and Western blotting, it was confirmed that dystrophin 1 and dystrophin 2 expression decreased and utrophin expression increased in the dystrophin mutant dog compared to the control. These results showed that, in addition to its resemblance to human clinical cases, the dystrophin mutant dog also exhibited histological lesions similar to dystrophinopathy patients. DMD is caused by out-of-frame mutations and the absence of the dystrophin protein in skeletal muscles because the dystrophin protein that is produced is truncated as a result of a premature stop codon and, therefore, is unstable [38]. BMD is caused by an in-frame mutation resulting in insufficient dystrophin protein, and clinical progression can be predicted by whether the deletion or duplication maintains or disrupts the translational reading frame [39,40]. However, there are reports that exceptions to the reading-frame rule occur in 10% of all DMD-causing mutations [31]. 

Our results show that the clinical characteristics of a dystrophin mutant dog are similar to pathologic features in human dystrophinopathy. Furthermore, a recent study reported that treatment with Cas9 and sgRNA-51 in spontaneous dystrophin KO dogs showed improved muscle histology [41]. Our dystrophin mutant dog will also be useful in research for developing therapeutics using the CRISPR/Cas9 system.

## 4. Materials and Methods

### 4.1. Ethics Statement

In this study, female mixed dogs from 2 to 4 years of age were used as oocyte donors and embryo transfer recipients. The dogs were housed indoors and fed once a day, with water provided ad libitum. For all experiments involving animals, methods and protocols were approved by the Committee for Accreditation of Laboratory Animal Care and the Guideline for the Care and Use of Laboratory Animals of Seoul National University (SNU-170310-14-1). All methods and protocols were carried out in accordance with the relevant guidelines and regulations.

### 4.2. Generation of CRISPR/Cas9

The pET plasmid (Appendix A) that encodes His-tagged Cas9 was transformed into BL21(DE3). Expression of Cas9 was induced using 0.5 mM IPTG for 4 h at 25 °C. The Cas9 protein was purified using Ni-NTA agarose resin (Qiagen, Germantown, MD, USA) and dialyzed against 20 mM HEPES (pH 7.5), 150 mM KCl, 1 mM DTT, and 10% glycerol. RNAs were in vitro transcribed through run-off reactions by T7 RNA polymerase. The template for sgRNA was generated by annealing and extension of two complementary oligonucleotides. Transcribed sgRNAs were preincubated with DNase I to remove template DNA and purified using PCR purification kits (Macrogen, Seoul, Korea). Purified RNA was quantified by NanoDrop spectrometry (Thermo Scientific, Wilmington, DE, USA). 

### 4.3. Cell Culture and Transfections 

Canine fetal fibroblasts were derived from the torso of a 27-day fetus post coitum and were cultured in DMEM (Gibco, Grand Island, NY, USA) with 10% FBS (Gibco). To introduce DSBs using an RNP complex, Cas9 protein was premixed with in vitro transcribed sgRNA and incubated for 10 min at room temperature. To introduce DSBs using plasmids, p3s-Cas9-2A-GFP (Appendix A) and pRG2-sgRNA (Appendix A) were premixed. Canine fetal fibroblasts were transfected with the Amaxa P3 Primary Cell 4D-Nucleofector Kit using Program EN-150 (Lonza, Basel, Switzerland). The transfected cells were transferred to 12-well plates, and further expansion was performed. Target mutations of cells were detected by T7 endonuclease 1 (T7E1) assay (New England Biolabs, Ipswich, MA, USA) (Appendix A). All primers are listed in Appendix A. The transfected cells were used as donor cells after verification of mutation frequency by T7E1 assay. Fibroblasts transiently expressing green fluorescence protein were collected from mixed cells and used as donor cells (Appendix A).

### 4.4. In Vivo Matured Oocyte Collection

After vaginal bleeding first appeared, blood was collected daily from the cephalic vein, and sera were separated by centrifuging for 10 min. Serum progesterone concentration was monitored by an IMMULITE 1000 (Siemens Healthcare Diagnostics Inc., Flanders, NJ, USA). The day when the progesterone concentration reached 4.0 ng/mL to 10.0 ng/mL was considered the day of ovulation. Seventy-two hours after ovulation, in vivo matured oocytes were collected by oviductal flushing using HEPES-buffered tissue culture medium-199 (TCM, Invitrogen, Carlsbad, CA, USA) supplemented with 10% bovine serum albumin and 2 mM NaHCO_3_. 

### 4.5. Somatic Cell Nuclear Transfer and Embryo Transfer

Cumulus cells were removed from in vivo matured oocytes by gentle pipetting in tissue culture medium-199 supplemented with 0.1% hyaluronidase. Metaphase chromosomes and extruded first polar bodies were removed under ultraviolet light by aspiration in HEPES-buffered TCM drops containing cytochalasin B and Hoechst 33442. Single donor cells were inserted into the perivitelline space of oocytes. Each donor cell–cytoplast couplet was fused with two pulses of DC 72 V for 15 μs using an Electro-Cell Fusion apparatus (NEPA GENE Co., Chiba, Japan). Fused embryos were activated in modified synthetic oviductal fluid (mSOF) medium containing 10 μM calcium ionophore (Sigma-Aldrich, St. Louis, MO, USA). After chemical activation for 4 min, cloned embryos were transferred to 40 μL of mSOF with 1.9 mM 6-dimethylaminopurine for 2 h. Reconstructed cloned embryos were transferred into the oviducts of synchronized recipients. Under laparotomy with general anesthesia, embryos were placed into the ampullary part of the oviduct using a 3.5 Fr Tom-Cat catheter (Sherwood, St. Louis, MO, USA).

### 4.6. T7 Endonuclease I Assay and Sequencing

Genomic DNA was isolated using a genome isolation kit (Promega, Madison, WI, USA) according to the manufacturer’s instructions. PCR amplicons were denatured at 95 °C, reannealed at 16 °C to form heteroduplex DNA using a thermal cycler, digested with 5 units of T7E1 assay for 20 min at 37 °C, and then analyzed using agarose gel electrophoresis.

### 4.7. Deep Sequencing Analysis

Genomic DNA was isolated from transfected cells and tail tissue of the cloned puppy. The target region was amplified using Phusion polymerase (New England Biolabs). Equal amounts of the PCR amplicons were subjected to paired-end read sequencing using Illumina MiSeq from Bio Medical Laboratories. Insertions or deletions located around the RGEN cleavage site (3 bp upstream of the PAM) were considered to be the mutations induced by RGENs.

### 4.8. Creatine Kinase

Blood samples for serum CK analysis were obtained from the jugular vein beginning at 10 days after birth. Samples were collected at 1-week intervals until 8 weeks. No attempt was made to limit exercise prior to sampling. Biochemistry analyses were performed on IDEXX Catalyst Dx (IDEXX VetLab Analysers, Westbrook, ME, USA). The normal reference range is 99–436 U/L.

### 4.9. Anesthesia for Magnetic Resonance Imaging

The dog was positioned in ventrodorsal recumbency and premedicated with 0.05 mg/kg glycopyrrolate (Mobinul, Myungmoon Pharmaceutical Co., Seoul, Korea). Anesthesia was induced with 0.06 mg/kg propofol (Provive, Myungmoon Pharmaceutical Co., Seoul, Korea) intravenously and was maintained with 1.5% isoflurane (Foran solution, Choongwae Pharma Corporation, Seoul, Korea) in 100% oxygen by endotracheal intubation. Heart rate, respiratory rate, and end-tidal CO_2_ concentration were monitored (MRI Patient Monitor, GE Medical System, Milwaukee, WI, USA).

### 4.10. Non-Quantitative (Conventional) Magnetic Resonance Imaging

MRI exams were performed with a 1.5-Tesla scanner (Signa HDx, GE Medical Systems, Milwaukee, WI, USA) using an 8-channel receiver coil (HD Knee PA coil, GE Medical Systems). MRI acquisition planes and parameters are summarized in Appendix A. T1-weighted images and T2-weighted images with fat suppression were acquired for anatomic reference and to assess damage to the hindlimb muscles, respectively. Gadolinium 0.1 mmol/kg (Magnevist, Bayer Korea, Seoul, Korea) was used intravenously for contrast study of the T1-weighted sequence.

### 4.11. Quantitative (T2 Mapping) Magnetic Resonance Imaging

Axial T2 maps were obtained by using multiecho sequences. Imaging parameters were follows: FOV, 18 × 18 mm; slice thickness, 5 mm; interslice gap, 0.3 mm; relaxation time, 1000 ms; echo time (ms), 7.4, 14.8, 22.2, 29.6, 37, 44.4, 51.7, and 59.1; matrix, 256 × 192; Nex, 2. Two radiologists (J.H. Kim and K.D. Eom) calculated T2 values of each hindlimb muscle at the mid-femur level. The size of ROI was fixed at approximately 5 mm^2^ considering individual muscle size. Mean and standard deviation of T2 values of each of the following seven muscles were acquired: rectus femoris, vastus lateralis, vastus medialis, biceps femoris, semitendinosus, semimembranosus and adductor magnus.

### 4.12. Biopsy and Histopathological Analysis

For biopsies of biceps femoris muscles, the dystrophin mutant dog and the control dog were anesthetized with ketamine and xylazine via intravenous injection, and anesthesia was maintained with isoflurane. The dogs were positioned in left lateral recumbency, and the biopsy region was prepared aseptically. After a 3 cm skin incision, a sample of the right biceps femoris muscle (1 cm × 1 cm × 0.5 cm) was collected from each dog. Immediately, the biopsies were flash-frozen in isopentane precooled in liquid nitrogen. A standard panel of histochemical stains and reactions was performed on 5 μm muscle cryosections. Additional cryosections were used for immunohistochemical staining using monoclonal antibodies against the dystrophin carboxy-terminal domain, rod domain, and utrophin (Novocastra, Newcastle-upon-Tyne, UK).

### 4.13. Western Blotting

Proteins of the dystrophin mutant dog and the control dog were extracted from each muscle sample. After measuring protein concentration, equal amounts of proteins were resolved by 10% SDS-PAGE and transferred to nitrocellulose membranes (Hybond; Amersham Biosciences, Amersham, UK). Membranes were blocked for 1 h in Tris-buffered saline Tween (TBST) containing 5% powdered skim milk and incubated overnight with the following primary antibodies: anti-dystrophin NCL-DYS1 (1:500, Novocastra Laboratories, Newcastle, UK), anti-dystrophin NCL-DYS2 (1:100, Novocastra Laboratories), and anti-utrophin NCL-DRP (1:100, Novocastra Laboratories). Horseradish peroxidase (HRP)-conjugated secondary antibody (1:3000, Santa Cruz Biotechnologies, Piscataway, NJ, USA) was used to detect bound antibodies with the Imaging System from FUSION-Solo (6×, Vilber Lourmat, Collégien, France).

## 5. Conclusions

Our study demonstrated that donor cells with CRISPR/Cas9 for a specific gene combined with the SCNT technique could efficiently produce a dystrophin mutant dog. Furthermore, this dystrophin mutant dog showed features such as CK elevation, dystrophin deficiency, skeletal muscle defects, abnormal ECG, and avoidance of ambulation. In the future, we hope that canine dystrophinopathy models with an in-frame dystrophin mutation will help in the successful development of new exon-skipping drugs.

## Figures and Tables

**Figure 1 ijms-23-02898-f001:**
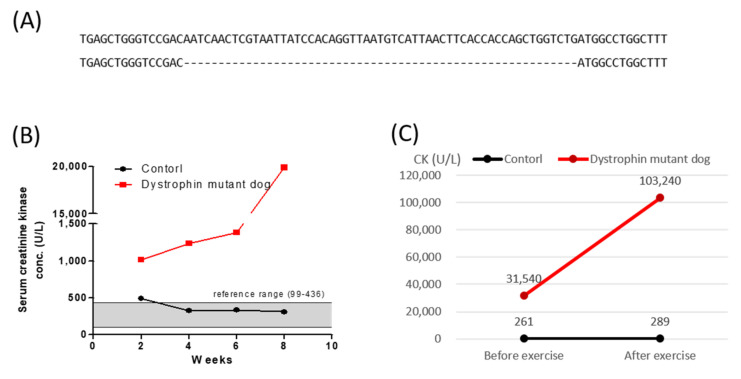
Production of a dystrophin mutant cloned dog by Cas9/sgRNA. (**A**) Sequences of target dystrophin locus detected in a cloned dog. The cloned puppy had a 57 bp deletion, and deleted sequences are shown. (**B**) Summary of serum creatine kinase (CK) values from two to eight weeks of age in the control and dystrophin mutant dog. (**C**) Change in creatine kinase in normal and dystrophin mutant dogs after exercise.

**Figure 2 ijms-23-02898-f002:**
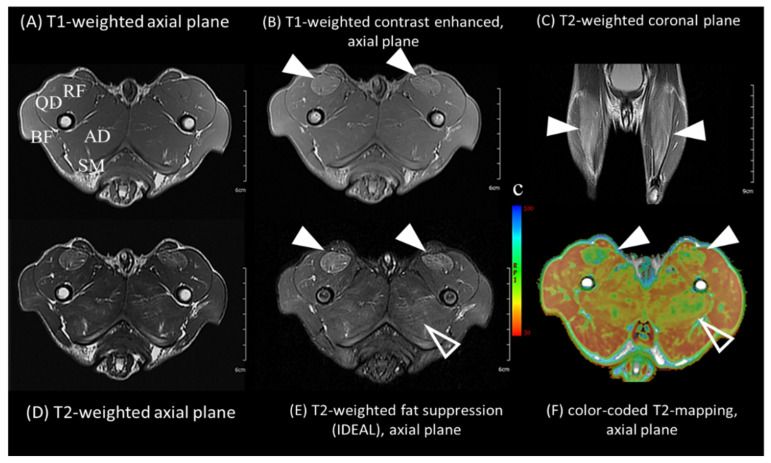
Magnetic resonance imaging of the dystrophin mutant dog. Rectus femoris (RF, arrowheads) and adductor magnus (AD, open arrowheads) muscles show marked hyperintense, contrast-enhancing lesions with minimal fatty replacement. On T1-weighted image (**A**,**B**), quadriceps muscles (QD) are moderately atrophied. Note diffuse lesions are found in anterior-medial hindlimb muscles with low T2 values (green color) on (**F**). Posterior-lateral hindlimb muscles, including biceps femoris (BF), semitendinosus (ST), and semimembranosus (SM) muscles, tend to be relatively uninvolved.

**Figure 3 ijms-23-02898-f003:**
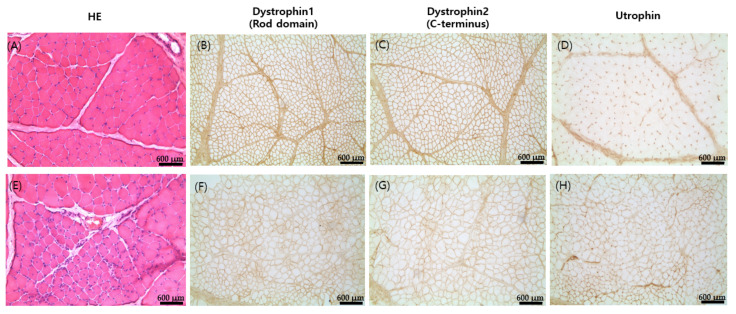
Histopathological analyses of dystrophic muscle in control (**A**–**D**) and dystrophin mutant dog (**E**–**H**). Muscle pathology showing focal necrosis and regeneration of muscle fibers (HE); immunohistochemical staining using monoclonal antibody against dystrophin rod domain and utrophin showing decreases in dystrophin 1 and dystrophin 2 expression and increased utrophin expression compared to control muscles (scale bar = 600 µm).

**Figure 4 ijms-23-02898-f004:**
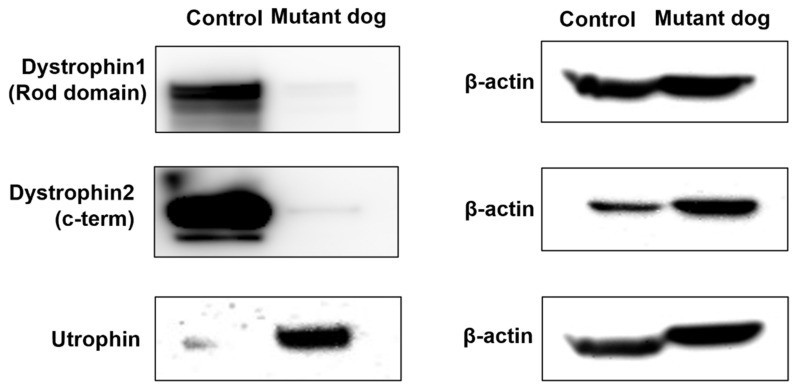
Western blot confirming the negligible expression of dystrophin 1 and 2 and upregulation of utrophin in dystrophin mutant dog (Mutant dog) and control dog.

**Table 1 ijms-23-02898-t001:** Summary of embryo transfer and generation of the dystrophin mutant cloned pup.

Recipient	No. In Vivo Matured Oocytes	No. Reconstructed Oocytes	No. Transferred Embryos	Pregnancy	No. Births
A	16	6	6	+	1
B	13	7	7	-	0
C	20	13	13	-	0
Total	49	26	26	1 (33.3%) ^∫^	1 (3.84%) ^∫∫^

^∫^ The percentage is based on the total number of recipient dogs. ^∫∫^ The percentage is based on the total number of transferred embryos.

## Data Availability

All data generated or analyzed are available from the corresponding author on request.

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
