# Peer review of "Generation of a Dystrophin Mutant in Dog by Nuclear Transfer Using CRISPR/Cas9-Mediated Somatic Cells: A Preliminary Study"

_ijms, 2022, doi:10.3390/ijms23052898_

Round 1

Reviewer 1 Report

25th February, 2022

Review of the Manuscript ID: ijms-1618897, by H.J. Oh et al., entitled: “Generation of a dystrophin mutant in dog by nuclear transfer using CRISPR/Cas9-mediated somatic cells” that is intended to be published as the Article in International Journal of Molecular Sciences

(separate Microsoft Word file as Reviewer Attachment for Manuscript ID ijms-1618897 Int. J. Mol. Sci. 25th February 2022 that includes Comments to the Authors is also uploaded)

Taking into account research highlight, contribution of the Authors to the progress in the research area, thorough manner of data presentation, very well writing in English, abundance of Materials and Methods and Results, and diligent tabular and graphic/photographic documentation, the quality of this paper deserves praise and merits my support. The Authors have received the tremendously high scores from me for the originality, importance of the work and the scientific value of their paper. In my opinion, the current paper provides insightful interpretation of topical and coming trends in devising and optimizing the strategy of successful generation of somatic cell-cloned dogs that display CRISPR/Cas9-mediated knockout of dystrophin gene. These dystrophin-deficient cloned dogs can provide canine transgenic model that can be reliable and feasible for not only comprehensive exploration of etiopathogenesis and molecular nature of human dystrophinopathy but also development of efficient clinical treatments in patients afflicted with such X-linked recessive muscular dystrophies as Duchenne muscular dystrophy (DMD) and Becker muscular dystrophy (BMD).

For all the above-indicated reasons, I highly recommend the Editorial Board to allow for publication of this excellent paper in International Journal of Molecular Sciences, after the minor revision of the manuscript will have been completed by the Authors and provided that the Authors are ready to consider all the Reviewer comments shown below:

1) There is a lack of the separate Conclusions section at the end of manuscript. Therefore, the summary part of the Discussion should have been transferred to the separate Conclusions section.

2) There is a lack of the Abbreviations section in the paper. That is why, this section should have been added by the Authors. The Abbreviations section should have been prepared in order to thoroughly elucidate and expand a broad spectrum of the in-text abbreviations, which have been used by the Authors in all the sections of their paper.

General Comment of the Reviewer:

Before the manuscript will have been accepted for publication in International Journal of Molecular Sciences, it requires the minor revision (according to all the recommendations indicated above by the Reviewer).

Author Response

Thank you for the reviewer’s comments. All the details pointed out by the reviewer greatly contributed to improving our manuscript. We substantially revised the manuscript and provided the list of changes made as follows.

Q1) There is a lack of the separate Conclusions section at the end of manuscript. Therefore, the summary part of the Discussion should have been transferred to the separate Conclusions section.

A1) As pointed, we added Conclusions section at the end of manuscript as follows;

  1. Conclusions

Our study demonstrated that donor cells with CRISPR/Cas9 for a specific gene com-bined with the SCNT technique could efficiently produce a dystrophin mutant dog. Fur-thermore, this dystrophin mutant dog showed many features such as CK elevation, dys-trophin deficiency, skeletal muscle defects, abnormal ECG, and avoidance of ambulation. In the future, we hope that canine dystrophinopathy models with an in-frame dystrophin mutation will help in the successful development of new exon-skipping drugs.

Q2) There is a lack of the Abbreviations section in the paper. That is why, this section should have been added by the Authors. The Abbreviations section should have been prepared in order to thoroughly elucidate and expand a broad spectrum of the in-text abbreviations, which have been used by the Authors in all the sections of their paper.

A2) As pointed, we added Abbreviations section in References as follows;

Abbreviations: Duchenne muscular dystrophy (DMD), Becker muscular dystrophy (BMD), gold-en retriever muscular dystrophy (GRMD), canine X-linked muscular dystrophy (CXMD), knockout (KO), creatine kinase (CK), electrocardiogram (ECG), magnetic resonance imaging (MRI), somatic cell nuclear transfer (SCNT), clustered regularly interspaced short palindromic repeats/CRISPR-associated system (CRISPR/Cas9)

Reviewer 2 Report

in this work, Ju Oh et al have demonstrated that dystrophin mutant dog could be generated by nuclear transfer using donor cells in which the dystrophin gene is knocked out by the CRISPR/Cas9 system. the overall work is sound and the approach is acceptable, however, the main weakness is having only one dog in the test group which was compared to the control.

This should be highlighted in the title as "a Preliminary Study"

the quality of the western blot is not acceptable and the whole blots should be presented.

the authors should present the sequence of the plasmid and show the cell morphology that was transfected etc.. ie highlighting the methodology approach.

Minor:

please correct the sentence "occurring animal 12models are available to" in the introduction

Author Response

Thank you for the reviewer’s comments. All the details pointed out by the reviewer greatly contributed to improving our manuscript. We substantially revised the manuscript and provided the list of changes made as follows.

Author’s corrections

  1. We removed the abbreviation on line 28 in the abstract and wrote SCNT as somatic cell nuclear transfer.
  2. In line 128, we changed "the muscle aagnetic resonance imaging" to the "muscle magnetic resonance imaging".
  3. The first abbreviation of MRI was placed on line 129.
  4. We changed magnetic resonance imaging to MRI in line 190.
  5. We changed the content of the paragraphs in lines 232-234 as follows:
    Our results showed that the clinical characteristics of a dystrophin mutant dog are similar to the pathologic features in human dystrophinopathy. Furthermore, a recent study reported that treatment with Cas9 and sgRNA-51 in spontaneous dystrophin KO dogs showed improved muscle histology [41]. Our dystrophin mutant dog will also be useful in research for developing therapeutics using the CRISPR/Cas9 system.
  6. We changed “he” to “the” in line 342.